# Genome-Wide Analysis of the *TCP* Transcription Factor Gene Family in Pepper (*Capsicum annuum* L.)

**DOI:** 10.3390/plants13050641

**Published:** 2024-02-26

**Authors:** Zeyu Dong, Yupeng Hao, Yongyan Zhao, Wenchen Tang, Xueqiang Wang, Jun Li, Luyao Wang, Yan Hu, Xueying Guan, Fenglin Gu, Ziji Liu, Zhiyuan Zhang

**Affiliations:** 1Hainan Institute, Zhejiang University, Sanya 572000, China; 22016141@zju.edu.cn (Z.D.); 22016137@zju.edu.cn (Y.H.); 22016144@zju.edu.cn (Y.Z.); tangwenchen1999@163.com (W.T.); wangxueqiang02@163.com (X.W.); lijun20181204@zju.edu.cn (J.L.); 2171100066@zju.edu.cn (L.W.); 0016211@zju.edu.cn (Y.H.); xueyingguan@zju.edu.cn (X.G.); 2Hainan Key Laboratory for Biosafety Monitoring and Molecular Breeding in Off-Season Reproduction Regions, Sanya 572000, China; 3Sanya Research Institute, Chinese Academy of Tropical Agricultural Sciences, Sanya 572000, China; 4Tropical Crops Genetic Resources Institute, Chinese Academy of Tropical Agricultural Sciences, Haikou 571101, China; 5Key Laboratory of Crop Gene Resources and Germplasm Enhancement in Southern China, Ministry of Agriculture, Haikou 571101, China

**Keywords:** *Capsicum annuum* L., TCP transcription factors, shoot branching, hormone response, abiotic stress

## Abstract

TCP transcription factors play a key role in regulating various developmental processes, particularly in shoot branching, flower development, and leaf development, and these factors are exclusively found in plants. However, comprehensive studies investigating TCP transcription factors in pepper (*Capsicum annuum* L.) are lacking. In this study, we identified 27 *CaTCP* members in the pepper genome, which were classified into Class I and Class II through phylogenetic analysis. The motif analysis revealed that *CaTCPs* in the same class exhibit similar numbers and distributions of motifs. We predicted that 37 previously reported miRNAs target 19 *CaTCPs*. The expression levels of *CaTCPs* varied in various tissues and growth stages. Specifically, *CaTCP16*, a member of Class II (CIN), exhibited significantly high expression in flowers. Class I *CaTCPs* exhibited high expression levels in leaves, while Class II *CaTCPs* showed high expression in lateral branches, especially in the CYC/TB1 subclass. The expression profile suggests that *CaTCPs* play specific roles in the developmental processes of pepper. We provide a theoretical basis that will assist in further functional validation of the *CaTCPs*.

## 1. Introduction

The transcription factor (TF) is an essential protein that can bind to specific DNA sites, playing a pivotal role in regulating gene expression levels [1], including plant morphogenesis [2], the cellular life cycle [3], and responding to abiotic stresses [4]. The *TCP* gene family is named after the initials of its three members, TEOSINTE BRANCHED1 (TB1), CYCLOIDEA (CYC), and PROLIFERATING CELL FACTORS 1 and 2 (PCF1 and PCF2). TB1 is involved in maintaining apical dominance in maize (*Zea mays*) [5], CYC regulates floral symmetry in snapdragon (*Antirrhinum majus*) [6], and PCF functions in the cell cycle of rice (*Oryza sativa*) [7].

TCPs share a highly conserved non-canonical structure known as the basic-helix-loop-helix (bHLH) motif, which consists of 59 amino acids at the N-terminus. The special structure was called the TCP domain [2]. The functions of the domain include binding to DNA sites and participating in various protein interactions [8]. The TCP family is composed of two classes: Class I, which contains the PCF class [9,10], and Class II, which is further classified into two subclasses, CYC/TB1 and CIN, based on the dissimilarity of the TCP domain [11,12,13]. Additionally, members of Class II contain an arginine-enriched motif consisting of 18–20 amino acid residues. This motif is called the R domain, and it may arise from specific secondary structural proteins, resulting in participation in protein interactions [7].

Previous studies have confirmed that *TCPs* regulate various growth and development processes in plants, including embryonic growth [14], branching growth [15], floral symmetry [16], internode length [17], and leaf development [18,19]. The mechanism of *TCPs* responding to various hormones and abiotic stresses has been explored. For instance, in *Arabidopsis*, *TCP9* and *TCP19* have been shown to promote leaf senescence in response to treatment with jasmonic acid (JA). *TCP14* promotes the process of *Arabidopsis* seed germination by responding to abscisic acid (ABA) signaling [20]. In *Arabidopsis* flowers, gynoecium and silique developmental processes are modulated by *TCP15* through regulating auxin biosynthesis [21]. Furthermore, *TCPs* play key roles in regulating various phytohormone signaling, including salicylic acid (SA), brassinosteroids (BRs), strigolactones (SLs), and gibberellic acid (GA) [22]. *OsTCP19* responds to abiotic stresses by regulating the expression level of *ABI4* [23], and the overexpression of *OsTCP14* contributes to enhancing rice cold tolerance [24]. In cotton, the expression of *GhTCPs* was significantly upregulated in responding to drought, heat, and salt stresses [25]. *TCPs* exhibit varying expression levels in different organs, and many *TCPs* have been found to show widespread and less tissue-specific expression profiling, such as in flowers, leaves, buds, and fruits in grapevine [26] and cassava [27]. In a nutshell, *TCPs* responded to abiotic stress and hormone signaling, and participated in various growth and developmental processes with diverse biological functions [11].

With the advancement of genome technologies, the number of plant species in which the *TCP* gene family has been identified is gradually increasing. For example, 22 *TCPs* in *Oryza sativa* [28], 24 *TCPs* in *Arabidopsis*, 38 *TCPs* in *Gossypium raimondii* L. [29], 73 *TCPs* in allotetraploid cotton (*Gossypium barbadense* L.) [25], 30 *TCPs* in tomato (*Solanum lycopersicum*) [30], and 31 *TCPs* in potato (*Solanum tuberosum* L.) had been analyzed [31]. Pepper (*Capsicum annuum* L.), a major worldwide spice crop of the Solanaceous, possesses great economic value as an ingredient for seasoning and medicine [32]. However, its growth, development, and productivity are sensitive to abiotic stress and various plant hormones [33,34]. The pepper genome was reported in 2014 [32,35]. However, the identification of *TCP* gene family members in pepper (*Capsicum annuum* L.) has not yet been conducted. To gain further insights into *TCPs* in pepper, we conducted detailed analyses in the current research, including phylogenetic relationships, gene classification, synteny analysis, GO annotation, conservation motif studies, cis-element analysis, predictions of miRNA targeting sites, and the three-dimensional structure of *TCPs*. Furthermore, we assessed the expression profiling of *TCPs* in different organs and their response to various hormone signaling and abiotic stress. This research provides a theoretical foundation for further studies on the biological functions of *TCPs* in pepper.

## 2. Results

### 2.1. Identification and Characterization of TCP Family Members in Pepper

We performed a BlastP search against the pepper genome using known TCP protein sequences from model plants (*Arabidopsis* and rice) and closely related species of pepper (tomato and potato). The obtained sequences were further verified with HMMER search using Pfam domains: PF03634. Finally, we identified a total of 27 *TCPs* that contain the TCP domain in the pepper genome (Appendix A). The distribution of *CaTCPs* varied across different chromosomes. Chromosome 2 contained six *CaTCPs*, while chromosome 3 and chromosome 6 each had four *CaTCPs.* The remaining *CaTCPs* were distributed across chromosome 1, chromosome 5, chromosome 7, chromosome 8, chromosome 9, and chromosome 11 (Appendix A). The nucleotide lengths of the 27 *CaTCPs* ranged from 614 bp (*CaTCP24*) to 1649 bp (*CaTCP27*), while the amino acid lengths ranged from 204 (*CaTCP24*) to 549 aa (*CaTCP27*) (Appendix A). Out of the 27 CaTCP proteins, 26 were predicted to be located in the nucleus, while one CaTCP protein was found in the cytoplasm (Appendix A).

### 2.2. Phylogenetic Analysis and Classification of CaTCPs

We constructed an unrooted phylogenetic tree (Figure 1) using 134 TCP proteins from *Capsicum annuum*, *Arabidopsis thaliana*, *Oryza sativa*, *Solanum tuberosum*, and *Solanum lycopersicum* to explore the evolutionary and phylogenetic relationships among these species. Following the standard classification in *Arabidopsis thaliana* [28], the 134 TCP members were divided into Class I (PCF) and Class II. Class I (PCF) comprises 62 TCP members (12 CaTCPs, 12 AtTCPs, 10 OsTCPs, 14 StTCPs, and 13 SlTCPs). Class II, which consists of 72 TCPs, was further divided into two subclasses: CYC/TB1 and CIN. Subclass CYC/TB1 includes 25 TCPs (5 CaTCPs, 3 AtTCPs, 4 OsTCPs, 7 StTCPs, and 6 SlTCPs), and CIN includes 47 TCPs (10 CaTCPs, 8 AtTCPs, 8 OsTCPs, 10 StTCPs, and 11 SlTCPs) (Figure 1). The tree revealed that most CaTCPs were phylogenetically closer to StTCPs and SlTCPs than to members from other species. The number of CaTCPs distributed in both subclasses is similar to the scattered distribution pattern of other species (Figure 1; Appendix A).

To determine the characteristic features of each class and subclass, we conducted an alignment analysis. We found 27 genes with highly shared TCP protein domains in pepper (Figure 2A). The 27 CaTCP protein sequences could be divided into 12 Class I (PCF) members and 15 Class II members, which include 5 CYC/TB1 subclass members and 10 CIN subclass members. In the basic motif, most members of Class I show a loss of four amino acids compared to Class II (Figure 2A). Within Class II, the Helix II motif of CYC/TB1 subclades lacks the amino acid “A” compared to the CIN subclass (Figure 2A). We observed that three members of Class II possess an R domain. These three members include two CIN members (*CaTCP5* and *CaTCP11*) and one CYC/TB1 member (*CaTCP15*) (Figure 2B). These findings are consistent with the results of previous phylogenetic analyses.

### 2.3. Chromosomal Location and Synteny Evaluation of TCP Genes

The pepper genome analysis revealed that 27 *CaTCPs* were unevenly distributed among 9 out of 11 pepper chromosomes (Figure 3). Chromosomes 2, 3, 6, and 8 exhibited a higher number of *CaTCPs*, with 6, 4, 4, and 3 genes, respectively, in contrast to chromosomes 1, 5, 7, 9, and 11, which harbored *CaTCPs* 1, 2, 2, 2, and 2, respectively. Chromosomes 4, 10, and 12 were devoid of *CaTCPs* (Figure 3). Remarkably, chromosomes 2 and 3 exhibited an enrichment in *CaTCPs*, with over 40% of *TCP* clusters on chromosomes 2 and 3. Despite accounting for only 5.81% (169.55 M) of the reference genome (2.85 G), chromosome 2 harbors 22.2% of the *CaTCPs*. Chromosome 3, which represents only 9.68% (282.78 M) of the pepper genome, contains 14.8% of the *CaTCPs*. Analysis of duplication events revealed the classification of *CaTCPs* into whole-genome duplication and tandem duplication (Appendix A). The analysis identified four putative paralog pairs of segmental duplication and two putative paralog pairs of tandem duplication (Figure 3). What can be seen in these results is that large-scale genome duplication events had a significant effect on the evolution of the *TCP* family in pepper.

Collinearity analysis was performed to uncover the evolutionary relationship among *TCPs* in different species (Figure 4; Appendix A). Chromosome 2 showed syntenic associations between 5 *CaTCPs*, 12 *StTCPs*, and 9 *SlTCPs*. Similarly, chromosome 3 exhibited syntenic relationships between 3 *CaTCPs* and 6 *StTCPs*, as well as 8 *SlTCPs*. (Figure 4; Appendix A). The results of collinearity analysis show that the evolutionary relationship of the *TCPs* in *Solanaceae* is complex, especially in chromosome 2 and chromosome 3. On the other hand, the complex syntenic relationship between *Solanaceae* species shows that segmental or tandem duplication was an important factor of *CaTCP* expansion in the pepper genome (Figure 4; Appendix A).

The Ka/Ks ratio is commonly employed to investigate the selection pressures acting on sequences. Generally, Ka/Ks > 1 indicates that the gene is under strong positive selection during the process of evolution, On the contrary, Ka/Ks < 1 indicates that the gene is under purifying selection. Consequently, Ka/Ks ratios of *TCP* genes were calculated among the four species to explore the evolutionary processes of *CaTCPs* (Appendix A). The results show that the Ka/Ks ratio of the duplicated *CaTCPs* pairs is less than 1 (Appendix A), indicating that some *CaTCPs* may be lost owing to selective pressure (Appendix A). The selection pressure analysis results were consistent among *Arabidopsis thaliana*, *Oryza sativa*, *Solanum tuberosum*, and *Solanum lycopersicum* (Appendix A).

### 2.4. Assessment of Gene Structures and Conserved Motifs, and Recognition Sequence of miR319

Analysis of the exon/intron structure and configuration of *CaTCPs* was conducted to gain deeper insights into the diversification of *CaTCPs* (Figure 5C). The gene structure of *CaTCPs* was analyzed by aligning the coding sequence (CDS) of each *CaTCP* gene with the corresponding pepper genomic sequences. The results revealed that four out of twelve *CaTCPs* in Class I (PCF) contained introns. The CIN subclass of *CaTCPs* exhibited a conserved gene structure, with nine out of ten *CaTCPs* lacking introns, while *CaTCP3* possessed a single intron (Figure 5C). Within the CYC/TB1 subclass, three out of five *CaTCPs* contained introns (Figure 5C). The exon–intron arrangement of *TCP* genes was comparatively constant, especially *CaTCP*s of the same class, which retained highly consistent gene structures (Figure 5C).

The motifs of *CaTCPs* were analyzed by identifying conserved regions within their protein sequences, providing a deeper understanding of the evolutionary relationship among *CaTCPs* (Figure 5B). In total, we predicted 20 motifs (Appendix A). The number of conserved motifs among the *CaTCPs* ranged from 5 to 13 (Appendix A). As expected, all 27 *CaTCPs* exhibited a highly conserved TCP domain. Motifs 1, 2, and 3 were present and conserved in all *CaTCPs*. Motifs 17 and 19 were only present in Class I. In short, motif structures and the distribution of CaTCP proteins were consistent with the classification of the class, suggesting that *TCPs* within the same subclass may share similar biological functions.

In *Arabidopsis*, miR319a controls leaf senescence and JA biosynthesis by binding to *TCP* transcription factors. The *TCPs* that can be combined by *AtmiR319* include *AtTCP2*, *AtTCP3*, *AtTCP4*, *AtTCP10*, and *AtTCP24* which belong to Class II [18]. In *Capsicum annuum*, the evolutionarily closest homologs of the *Arabidopsis* genes are *CaTCP2*, *CaTCP4*, *CaTCP5*, *CaTCP19*, and *CaTCP26*, which can be bound by miR319. These homologs show closer proximity to the *AtTCPs* in the phylogenetic tree (Figure 1 and Figure 2B). The results suggest that *miR319* probably plays a key role in regulating pepper development by binding to these *CaTCPs* (Figure 5C; Appendix A).

### 2.5. Genome-Wide Prediction of miRNA Targeting CaTCPs

Over the past decade, an increasing body of literature has emphasized the involvement of miRNA binding to target genes in abiotic and biotic stress responses. Hence, to boost our understanding of miRNAs linked to the regulation of *CaTCPs*, we predicted 37 miRNAs targeting 19 *CaTCPs* (Figure 6; Appendix A). *CaTCP23*, *CaTCP16,* and *CaTCP19* were forecasted to be regulated by a larger number (respectively, 15, 11, 8) of miRNAs (Figure 6; Appendix A). It is interesting that both miR319 and miR159 can target five *CaTCPs* (Appendix A). To explore the biological function of the miRNAs and targeted *CaTCPs*, additional research validating their expression profiling is required.

### 2.6. GO Annotation and Enrichment Analysis of CaTCPs

GO annotation and enrichment analysis was used to further study the biological functions of *CaTCPs*. The analysis included biological process (BP), molecular function (MF), and cellular component (CC) classes. Several suggestively enriched terms were identified and presented (Appendix A). For example, in the BP enrichment analysis class, 19 principally enriched terms were uncovered, involving the regulation of macromolecule metabolic processes (GO:0060255), metabolic processes (GO:0019222), and cellular processes (GO:0050794) (Appendix A). In the CC enrichment analysis, we identified ten primarily enriched terms, involving intracellular membrane-bound organelles (GO:0043231), obsolete cell parts (GO:0044464), membrane-bound organelles (GO:0043227), and obsolete cells (GO:0005623) (Appendix A). Outcomes of the MF class distinguished eight highly enriched terms linking with molecular function (GO:0003674), and organic cyclic compound binding (GO:0097159) (Appendix A). In short, the GO enrichment analysis confirmed the involvement of *CaTCPs* in DNA-templated transcription, heterocyclic compound binding, and the transcriptional regulation of pepper growth and development stages.

### 2.7. Expression Profiling of CaTCPs in Different Organs and Development Stages

We determined the tissue-specific expression levels of *CaTCPs* in five major tissues: root, stem, leaf, placenta, and pericarp (Figure 7). The relative transcript abundance of *CaTCPs* revealed distinct expression patterns for each gene across different tissues and developmental stages. For instance, *CaTCP16* exhibited higher expression levels in flowers but lower expression levels in roots and stems (Figure 7). The expression levels of the genes varied across developmental stages. For instance, in the placenta, the expression of *CaTCP7* gradually increased over the development period, while the expression of *CaTCP25* gradually decreased (Figure 7). In the pericarp, the expression of *CaTCP1* gradually increased over the development period, whereas the expression of *CaTCP10* gradually decreased (Figure 7). These results suggest that certain genes exhibit unique expression profiles during different developmental stages. In conclusion, the RNA-seq data indicate that certain *CaTCPs* are likely involved in important biological functions that contribute to the growth and development of pepper.

To investigate the importance of *CaTCPs* in controlling leaf development and shoot branching, expression profiles were analyzed in stem leaves, branch leaves, flower buds, and lateral buds using qRT-PCR. All *CaTCPs* were expressed in the four tissues, but expression patterns varied between subclasses (Appendix A; Appendix A). Compared to Class II, most members of Class I had a special high expression in stem and branch leaves (Appendix A; Appendix A), especially with the high expression in branch leaves, indicating that Class I *CaTCPs* play a key role in controlling leaf development in pepper. In contrast, most *CaTCPs* belonging to Class II showed high expression in lateral branches, such as *CaTCP6*, *CaTCP8,* and *CaTCP17* (Appendix A; Appendix A). The results imply that Class II *CaTCPs* may play a key role in regulating shoot branching. Almost all members are expressed in flower organs, indicating that *CaTCPs* may have important regulatory functions in flower development. It was further determined that *CaTCPs* play specific roles in various aspects of pepper development and growth, especially in leaf growth and shoot branching. The results are based on the mining of publicly available transcriptome sequencing data, BioProject ID: PRJNA223222.

### 2.8. Expression Profiling Analysis of CaTCPs under Phytohormones and Abiotic Stress Conditions

The expression profiling of *CaTCPs* under abiotic (cold, heat, drought, and salt), and phytohormone [methyl jasmonate (MeJA), salicylic acid (SA), ethylene (ET), and abscisic acid (ABA)] treatments at different time points was conducted using public transcriptome data (Figure 8). In abiotic stress conditions, a few TCP genes had comparatively high expression levels compared to the control group. For example, *CaTCP8* was significantly upregulated under heat and cold stress. Notably, *CaTCP4*, which contains a low-temperature responsive element, exhibited significantly higher expression levels at 12 h under cold treatment (Figure 8A; Appendix A). Under drought stress, *CaTCP11* and *CaTCP26* showed significant upregulation (Figure 8A). Interestingly, under salt stress, three *CaTCP*s, *CaTCP1*, *CaTCP11*, and *CaTCP16*, had significantly higher expression levels at 6 h, 72 h, and 6 h, respectively. It is noteworthy that all these genes contain a drought-responsive element (Figure 8A; Appendix A).

Most genes showed upregulation under MeJA treatment. Notably, *CaTCP8*, which contains four MeJA responsiveness elements, exhibited higher expression (Figure 8B; Appendix A). In contrast, only three genes (*CaTCP11*, *CaTCP15*, and *CaTCP20*) showed comparatively high expression levels under ET treatment (Figure 8B). After SA treatment, *CaTCP2* and *CaTCP7*, which contain salicylic-acid-responsive elements, showed significantly higher expression levels (Figure 8B; Appendix A). After ABA treatment, few genes showed significant variation (Figure 8B). These results further validate the presence of cis-elements linked to phytohormones and abiotic stress in *CaTCPs*, indicating their specific physiological roles in response to signaling pathways and abiotic stress, particularly in the case of salt, MeJA, and SA treatment. The results are based on the mining of publicly available transcriptome sequencing data.

### 2.9. Prediction of Interaction Network of CaTCPs

To better understand protein interactional relationships between CaTCPs, we constructed an interaction network using the 27 CaTCPs. We filtered out interrelationships with confidence levels below 0.9, resulting in 22 interacting CaTCPs. The detailed data of the predicted interaction network for these 22 CaTCPs are recorded in Appendix A. The interaction network of CaTCPs revealed a complex relationship. CaTCP4 and CaTCP9 exhibited a co-expression phenomenon with CaTCP16, which show consistent changes in gene expression levels. CaTCP16 plays a vital role in the interaction network by directly interacting with 13 CaTCPs and indirectly interacting with 8 CaTCPs (Appendix A; Appendix A). CaTCP16 interacted with the regulation of developmental-process-related TCP genes, CaTCP3 and CaTCP4, and other CaTCPs, which had been annotated by GO as the regulation of developmental processes (GO:0050793) (Appendix A; Appendix A). CaTCP16 also interacted with shoot system development, CaTCP5, CaTCP9, and other CaTCPs. Therefore, CaTCP16 may be involved in cell differentiation and development rhythms in pepper.

### 2.10. Three-Dimensional Structure Prediction of CaTCPs Protein

To assess the 3D structure models of CaTCP proteins, we subjected the protein sequences to the Robetta server for prediction. We selected 22 high-quality structures based on their confidence score values. The results indicated that the predicted structures of *CaTCP* are highly reliable, with the most favored regions ranging from 82.1% to 91.4%, additional allowed regions ranging from 6.1% to 16.3%, generously allowed regions ranging from 0% to 1.7%, and disallowed regions ranging from 0% to 1.3% (Appendix A). The secondary structures of the CaTCP protein can be classified into four main classes: alpha helices, beta sheets, random coils, and extended strands. The distribution of secondary structures was as follows: alpha helices accounted for 8.06–36.03%, beta sheets accounted for 0.90–7.12%, random coils accounted for 42.20–74.41%, and extended strands accounted for 6.09–19.61% (Appendix A; Appendix A). The random coil was the predominant secondary structure, indicating a high degree of consistency in the secondary structures.

## 3. Discussion

### 3.1. Identification, Expansion, and Evolution of TCP Gene Family in Pepper

Pepper (*Capsicum annuum* L.) is the second-most widely cultivated vegetable in the *Solanaceae* family, following tomatoes [34]. TCP transcription factors have been reported to be involved in various processes [35,36,37,38,39]. To date, *TCPs* have not been identified in the genomes of unicellular algae. However, five to six *TCPs* have been found in basal land plants [40], and numerous members have been identified in gymnosperms and angiosperms [11,40]. In this study, we identified 27 *TCPs* in the pepper genome, including twelve *CaTCPs* in the genomic duplicated region, which were involved in whole-genome duplication (WGD) or tandem duplication. Tandem duplication is presumed to generate gene copies [41,42]. Another key mechanism for the expansion of gene families is whole-genome duplication (WGD). In the pepper genome, with the expansion and evolution of *CaTCPs* mainly owing to WGD, 29.6% of *CaTCPs* were involved in WGD (Appendix A); one possible reason for this could be that *Solanaceae* species have experienced additional WGD events [43,44].

### 3.2. CaTCP Expression Pattern during Various Tissue Growth Stages

Conducting expression analysis in various tissues and growth stages will enhance our understanding of the regulatory mechanisms and biological processes involving *CaTCPs* in pepper [45]. The expression of 27 *CaTCPs* was profiled in the flower, root, stem, placenta, and pericarp. The results revealed a significant high expression of *CaTCP7* in the placenta, *CaTCP12* in the stem, and *CaTCP16* in the flower. Several reports have confirmed that *TCPs* exhibit diverse expression levels as a result of their involvement in various biological processes [46,47,48]. For instance, in cotton, tissue-specific expression patterns of *GhTCP* genes in the root, stem, leaf, flower, fiber, and ovule were analyzed. In the current study, *CaTCP16* belonging to CYC/TB1 showed significantly high expression in flowers, suggesting a key role at this growth stage. In agreement with previous reports, our results suggest that CaTCPs in pepper have multiple roles at various stages of growth and development.

### 3.3. The Essential Role of TCPs in Shoot Branching

*TCPs* have been proven to play crucial roles in shoot branching, which is an important aspect of plant growth and development, influencing plant height, photosynthesis efficiency, and the transport of organic matter [38]. Previous extensive reports have highlighted the significance of the TCP gene family in regulating cell growth and proliferation in lateral branches. Specifically, members of the CYC/TB1 subclass can suppress the growth of lateral buds. For instance, in maize, cultivars that overexpress *tb1*, which belongs to the CYC/TB1 subclass, are given priority in artificial selection due to their contribution to the formation of strong apical dominance [5]. Similarly, the silencing of *SlBRC1*, a member of the CYC/TB1 subclass, leads to a decrease in apical dominance, dwarfing, and increased lateral branching. However, axillary buds located in the stem of tomato plants exhibit high expression levels of *SlBRC1*, resulting in the inhibition of outgrowth [49]. Peppers and tomatoes have similar branching patterns, such as sympodial growth and cymose inflorescences. CYC/TB1 members may have similar functions that inhibit shoot branching in pepper. Notably, *CaTCP6* and *CaTCP8*, which are homologous to *SlBRC1* and belong to the CYC/TB1 subclass, display high expression levels in lateral branches, and may play a crucial role in the establishment of apical dominance.

### 3.4. miRNA Participating in the Gene-Regulatory Mechanisms of Stress Response

MicroRNAs (miRNAs) are a diverse class of non-coding, single-stranded regulatory RNAs, typically consisting of 20–24 nucleotides [50,51,52]. These miRNAs play a key role in regulating gene expression by binding to complementary regions of target mRNAs [53]. Recent studies have validated that miRNAs control diverse cellular functions, including responses to various stresses and the regulation of growth processes in pepper plants [35,54]. In the current research, we predicted thirty-seven miRNAs targeting nineteen *CaTCPs*. These mRNA might be crucial players in the regulation of growth and stress response. Previous research supports this conclusion [18,24,55,56]. For instance, *TCP4* in *Arabidopsis* was targeted by miR319, contributing to the regulation of the cotyledon boundary and leaf serration formation and accelerating plant maturation [57,58]. *Arabidopsis* mutants lacking miR319 have a prolonged juvenile stage, indicating that miR319 plays a key role in the vegetative phase change [59]. These studies show that miRNAs play diverse roles in growth processes and responses to various abiotic stresses by regulating the expression pattern of targeted *CaTCPs*.

## 4. Materials and Methods

### 4.1. Identification and Characterization Analysis of the TCP Genes in Pepper

This study utilized the genome data of the pepper cultivar CM334. The genome sequences of *Capsicum annuum*, *Arabidopsis thaliana*, *Oryza sativa*, *Solanum tuberosum*, and *Solanum lycopersicum* were downloaded from Phytozome (https://phytozome-next.jgi.doe.gov/, accessed on 12 January 2022). Published TCP protein sequences, including 24 AtTCPs, 22 OsTCPs, 31 StTCPs, and 30 SlTCPs, were adopted to construct a local protein database with blast-2.5.0 using default parameters. Putative CaTCPs were determined using BLASTP against the local databases with default parameters. The Hidden Markov Model (HMM) file (PF03634) of the TCP domain was downloaded from the Pfam protein domain database (http://pfam.xfam.org/, accessed on 12 January 2022) to further verify the putative CaTCPs. HMMER 3.1 (http://www.hmmer.org/, accessed on 12 January 2022) was used to search CaTCPs with the e-value set to 1 × 10^−5^. Finally, 27 CaTCPs were the outcome of both BLASTP and HMMER. The subcellular localization of TCP proteins in the pepper was predicted via the WoLF PSORT server (https://wolfpsort.hgc.jp/, accessed on 12 January 2022).

### 4.2. Phylogenetics and Synteny Analysis of CaTCP Proteins

TCP protein sequences in *Arabidopsis*, rice, tomato, potato, and pepper were adopted to study the phylogenetic relationship. Initially, the software tool Mega X (https://megasoftware.net/home, accessed on 12 January 2022) was used to perform sequence alignments. Subsequently, we plotted the phylogenetic tree using the Neighbor-Joining (NJ) method with 1000 bootstrap replicates. Finally, the iTol (https://itol.embl.de/, accessed on 12 January 2022) website service was used to further display the tree. We used MCScanX (https://github.com/wyp1125/MCScanX, accessed on 12 January 2022) to explore the synteny relationships of *TCP* genes in *Arabidopsis*, tomato, potato, and pepper. Additionally, Ka/Ks values for all TCP gene pairs were calculated using TBtools [60].

### 4.3. Gene Structure and Conserved Motif Analysis

The pepper genome annotation file was downloaded from Phytozome. The structures of the CaTCPs genes were plotted using TBtools. The conserved motifs in CaTCP proteins were identified using Multiple Em for Motif Elucidation (MEME, https://meme-suite.org/, accessed on 12 January 2022) software using the following parameters: the motif width range was 6 to 13 and the maximum number of motifs was 20 [36].

### 4.4. Cis-Elements Analysis in CaTCP Promoters

We extracted 2000 bp of sequence upstream of the start codon of the CaTCPs as the promoter to explore the cis-acting binding elements. The cis-acting elements were predicted by matching sequences in the promoter with binding sites in the PlantCARE (http://bioinformatics.psb.ugent.be/webtools/plantcare/html/, accessed on 12 January 2022) database. Finally, the figure was plotted through TBtools v2.052 software.

### 4.5. Prediction of Putative miRNA Targeting CaTCPs and GO Annotation Analysis

The coding sequences of the 27 CaTCPs were used to search possible target miRNAs with the help of psRNATarget (https://bio.tools/psrnatarget#!, accessed on 12 January 2022). The uploaded miRNA sequences were previously reported [35]. We used Cytoscape software (V3.8.2; https://cytoscape.org/download.html, accessed on 12 January 2022) to display the interaction relationship between the targeted and related miRNAs. GO annotation analysis was performed with the help of the eggnog website (http://eggnog-mapper.embl.de/, accessed on 12 January 2022) and TBtools.

### 4.6. Transcriptomic Data Analysis of the CaTCPs in Diverse Tissues, Abiotic, and Hormone Conditions

The RNA-seq data (BioProject ID: PRJNA223222) include samples from the root, stem, leaf, placenta, and pericarp at various time points: 1 day, 2 days, 3 days, mature green (MG), breaker (B), 5 days post breaker, and 10 days post breaker in *Capsicum annuum* L. The breaker stage in fruit typically refers to the stage when the fruit begins to lose its green color and starts to show signs of ripening [32]. The analysis was performed using the CM334 reference genome. The raw sequence reads (BioProject ID: PRJNA525913) of CM334 under different conditions, including cold, man, NaCl, heat, and mock, were obtained from the NCBI. For the data, at the 6-true-leaf stage, the plants were exposed to temperatures of 10 °C and 40 °C to simulate cold and heat stress, respectively. For salinity stress, plants were subjected to treatment with 50 mL of a 400 mM NaCl solution; for osmotic stress, 50 mL of 400 mM mannitol was administered to the peppers. Additionally, the sequence data (PRJNA634831) of CM334 under treatments of abscisic acid (ABA), methyl jasmonate (MeJA), salicylic acid (SA), and ethylene (ET) were also downloaded from the NCBI. For the data, at the 6-true-leaf stage, pepper plants were sprayed with the following solutions on the underside of leaves: 5 mM sodium salicylate (SA), 100 μM methyl jasmonate (JA), 5 mM ethephon (ET), 100 μM (±)-ABA, or distilled water (mock). The Fragments Per Kilobase of the exon model per Million mapped reads (FPKM) were calculated through Hisat2 (v2.0.5) and Sringtie (v2.1.7) software. Fold-change was calculated. The expression levels were visualized using the R package pheatmap, based on log10 values.

### 4.7. RNA Isolation and Quantitative RT-PCR

Pepper plants (cultivar CM334) were grown in a walk-in greenhouse under 16 h light/8 h dark conditions at 25–28 °C. After six weeks, flower buds, lateral buds, stem leaves, and lateral leaves were collected from three different pepper plants for qRT-PCR. These tissue samples were frozen in liquid nitrogen and stored at −80 °C. RNA was extracted using a TIANGEN^TM^ RNA plant kit. cDNA was synthesized using an RNA Reverse Transcription Kit from Vazyme. The qRT-PCR was performed on the ABI 7500 real-time PCR system with SYBR-green dye from Vazyme. *CaUBI3* was an internal reference gene [33]. The 2^−ΔΔCT^ method was used to calculate the relative expression level. Primers for qRT-PCR are listed in Appendix A.

### 4.8. Prediction of Protein–Protein Interaction Network of CaTCPs

The 27 CaTCP proteins were submitted to the string website (https://string-db.org/, accessed on 12 January 2022) to predict the protein–protein interaction network of CaTCPs.

### 4.9. 3D Structure Prediction, Validation, and Visualization of CaTCP Proteins

We used the AlphaFold2 website (https://alphafold.ebi.ac.uk/, accessed on 12 January 2022) to predict the 3D structures of the CaTCP proteins. The quality of the putative models was checked using the structural analysis and verification server (SAVES) (http://services.mbi.ucla.edu/SAVES, accessed on 12 January 2022). The SOPMA (http://npsa-pbil.ibcp.fr/cgi-bin/npsa_automat.pl?page=npsa_sopma.html, accessed on 12 January 2022) website was used to analyze the secondary structure of the chosen models. Finally, we used PyMol to visualize the 3D structures of the proteins [37].

## 5. Conclusions

This study identified 27 *TCPs* in the pepper genome for the first time. These *CaTCP* members were unevenly distributed on 12 chromosomes. Genome-wide analysis was performed to gain further insights into *CaTCPs* in pepper, including *CaTCPs’* identification, gene phylogeny, sequence alignment, motif analysis, cis-element detection, miRNA prediction, GO enrichment analysis, the interaction network of CaTCPs proteins, and 3D structure prediction. Furthermore, the expression patterns of *CaTCPs* in different abiotic stresses and organs revealed their potential involvement in multiple key biological processes. In summary, the findings of this study make a significant contribution to the field by providing an important starting point for future investigations into the functions of *CaTCPs*. Specifically, further study of these genes can offer valuable insights into the growth and development of flowers, leaves, and shooting branches. Furthermore, these findings have important implications for developing effective pepper breeding strategies.

## Figures and Tables

**Figure 1 plants-13-00641-f001:**
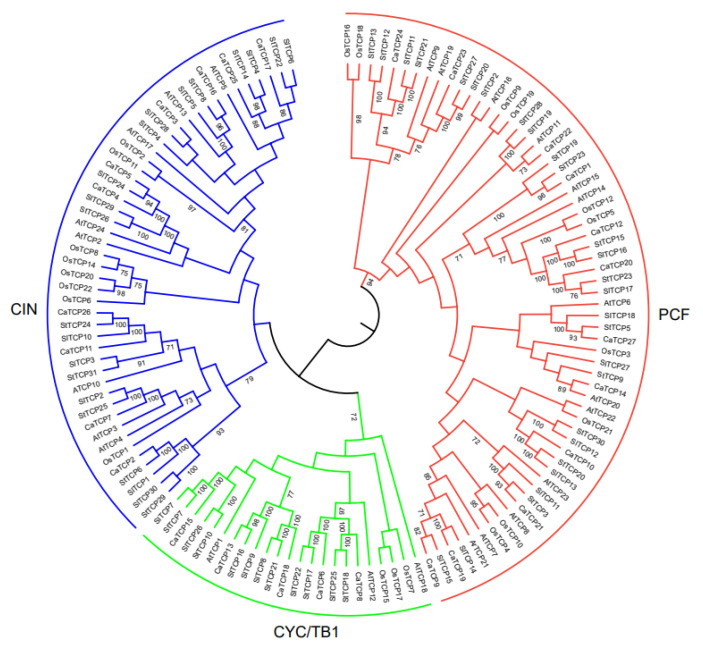
Phylogenetic relationship among TCP transcription factor families in *Capsicum annuum* (CaTCP), *Arabidopsis thaliana* (AtTCP), *Oryza sativa* (OsTCP), *Solanum tuberosum* (StTCP), and *Solanum lycopersicum* (SlTCP). The phylogenetic tree was constructed using the Neighbor-Joining method based on 134 full-length protein sequences from 27 CaTCPs, 24 AtTCPs, 22 OsTCPs, 31 StTCPs, and 30 SlTCPs. These TCP protein sequences were clustered into two major classes, Class I (red) and Class II (green, blue).

**Figure 2 plants-13-00641-f002:**
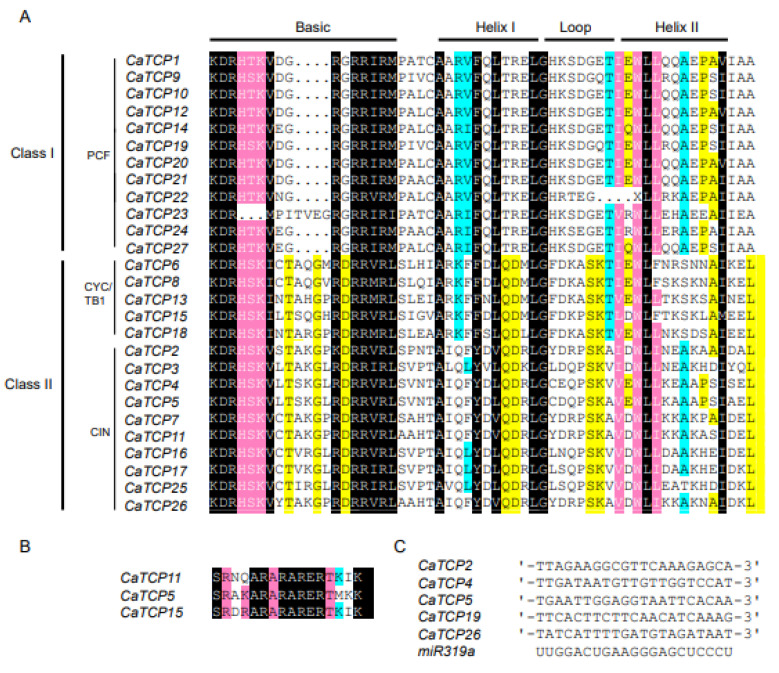
Multiple sequence alignment of two classes of CaTCP proteins. Conserved nucleotides are colored as follows: black, 100%; pink, 90–99%; cyan, 60–89%; and yellow, 50–59%. (**A**) The putative TCP domain for CaTCP proteins. (**B**) The putative R-domain for Class II subfamily members of CaTCP proteins. (**C**) Alignment of predicted target regions for miR319 complementary sequences.

**Figure 3 plants-13-00641-f003:**
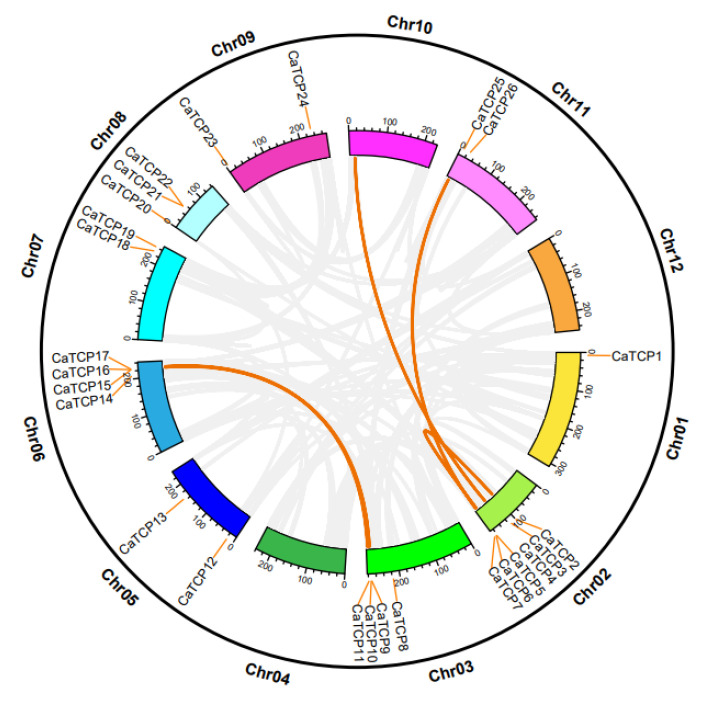
Chromosomal distribution and inter-chromosomal relation of the *CaTCPs*. Orange lines denote syntenic *CaTCP* gene pair blocks. All syntenic relationships in the pepper genome are indicated in gray.

**Figure 4 plants-13-00641-f004:**
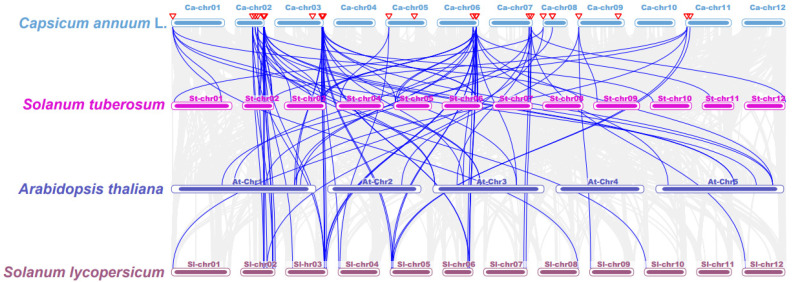
Synteny analysis of *TCP* genes between pepper, *Arabidopsis thaliana*, *Oryza sativa*, *Solanum tuberosum*, and *Solanum lycopersicum*. Grey lines present gene blocks that are orthologous to the other genomes. Blue lines indicate the syntenic *TCP* gene pairs. Red triangles indicate *CaTCPs*.

**Figure 5 plants-13-00641-f005:**
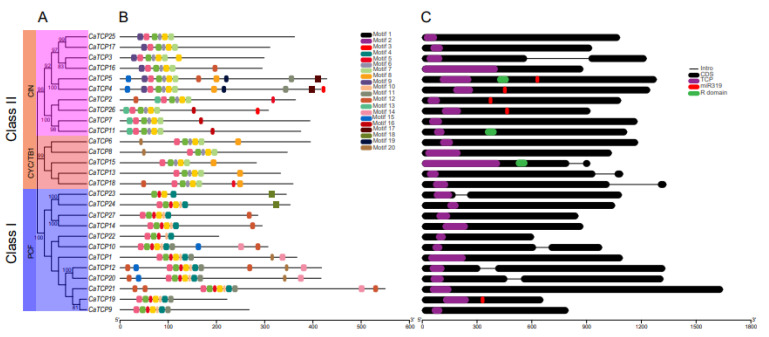
Phylogenetic tree, gene structure, and motif composition of *CaTCPs*. (**A**) Phylogenetic tree of CaTCP proteins. (**B**) Conserved motif compositions in CaTCP proteins. Each specific color has a distinct motif. (**C**) Exon and intron structure of *CaTCPs*. Exons and introns are shown by black rounded rectangles and black lines, respectively. Purple, red, and green rounded rectangles represent the TCP domain, miR319 recognition site, and R domain. The lengths of *CaTCPs* are indicated by the scale.

**Figure 6 plants-13-00641-f006:**
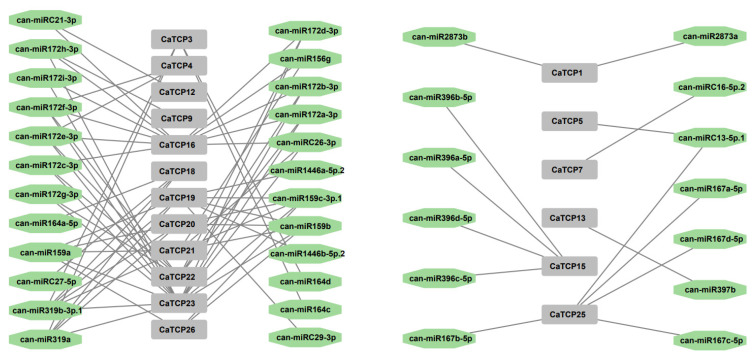
Network diagram of the regulatory linkages among the anticipated miRNAs and putative *CaTCPs*. Green octagon colors correspond to miRNAs, and gray rectangles represent *CaTCPs*.

**Figure 7 plants-13-00641-f007:**
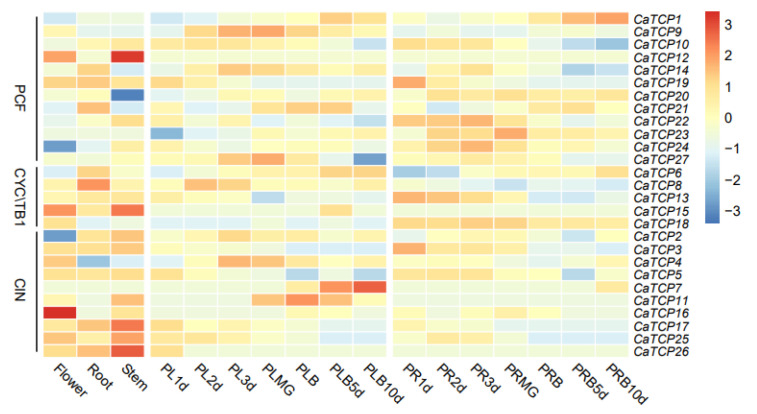
Expression pattern of *CaTCPs* in various tissues at different developmental stages. The different tissues of flower, root, stem, placenta, and pericarp. PL, PLMG, PLB, PR, and PRB tags represent the placenta, placenta green mature, placenta breaker, pericarp, and pericarp breaker stages of fruit development, respectively. The 1 d, 2 d, 3 d, 5 d, 10 d labels indicate the time points (days) at which the tissues were collected. High expression levels are indicated by the red color and low expression levels are indicated by the blue color. The vertical bar on the right shows the three groups of *CaTCPs*.

**Figure 8 plants-13-00641-f008:**
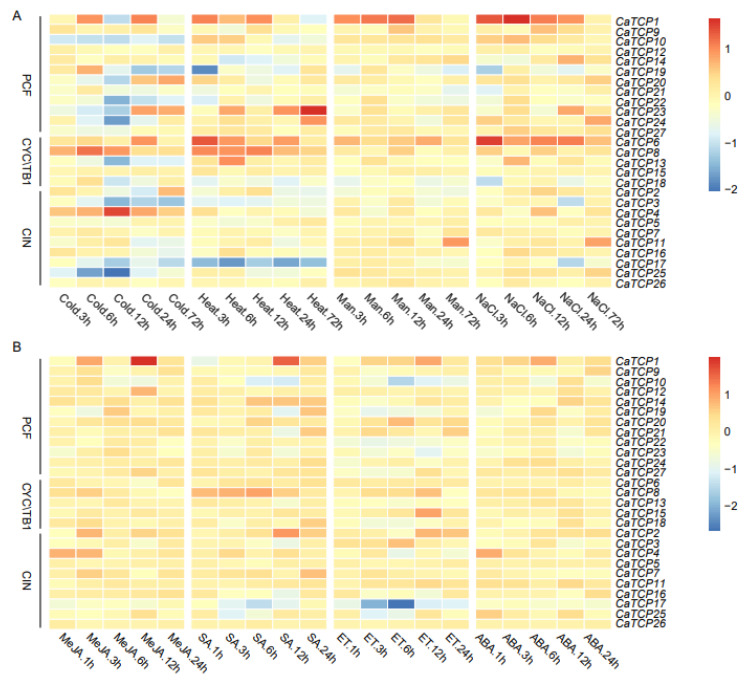
Expression profiling of CaTCPs under abiotic stress and phytohormone treatments. High expression levels are indicated by the red color and low expression levels are indicated by the blue color. The vertical bar on the right shows the three groups of CaTCPs. (**A**) Expression levels in seedings under abiotic stress conditions including cold, heat, drought (mannitol), and salt (NaCl) treatments for 3 h, 6 h, 12 h, 24 h, and 72 h. (**B**) Expression levels in response to phytohormone treatments including methyl jasmonate (MeJA), salicylic acid (SA), ethephon (ET), and abscisic acid (ABA) for 1 h, 3 h, 12 h, and 24 h. Numerous time points of abiotic stress contain 1 h, 3 h, 6 h, 12 h, 24 h.

## Data Availability

The original contributions presented in the study are included in the article/Appendix A, further inquiries can be directed to the corresponding author.

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
