# Peer review of "Genome-Wide Analysis of the TCP Transcription Factor Gene Family in Pepper (Capsicum annuum L.)"

_plants, 2024, doi:10.3390/plants13050641_

Round 1

Reviewer 1 Report

Comments and Suggestions for Authors

Article entitled “Genome-Wide …. Pepper” deals with in-silico analysis of TCP transcription factor in pepper genome. Authors identified 27 TCP and analyzed them further. TCP-TFs expressed differentially and play a major role in the developmental process of the plant.

The study is good and performed in logical manner. Results are supporting and sufficient enough for the publication.

Reviewer 2 Report

Comments and Suggestions for Authors

I checked your manuscript and described comments below.

Capsicum annuum L. is a globally important spice from the Solanaceae family.

This paper provides a very good analysis of the TCP gene of Capsicum annuum L.

I think you should consider the following points.

1.       MGEA X and HMMER 3.1 are old software. Analysis should be performed with the latest MEGA 11 or HMMER 3.4.

2.       Supplementary files have not been uploaded. It is referenced in the text, but the data cannot be verified.

I don't think this paper has new various major mistakes or grammatical problems.

Reviewer 3 Report

Comments and Suggestions for Authors

Title: Genome-Wide Analysis of the TCP Gene Family in Pepper (Capsicum annuum L.)

TCP transcription factors play a key role in regulating various developmental processes, particularly in shoot branching, flower development and leaf development, and these factors are exclusively found in plants. The TCP gene family, named after the initials of its three members, TEOSINTE BRANCHED1 (TB1), CYCLOIDEA (CYC), and PROLIFERATING CELL FACTORS 1 and 2 (PCF1 and PCF2). The TCP gene family is classified as class I (PCF), and class II (CYC/TB1 and CIN) based on the dissimilarity of the TCP domain. Considering the importance of its role in developmental processes, especially in shoot branching, flower development and leaf development, these TCP gen family were genome wide characterized in many crops. However, comprehensive studies investigating TCP transcription factors in pepper (Capsicum annuum L.) are lacking.

Therefore, in this study, 27 CaTCP members were identified in the pepper genome, which were classified into Class I and Class II through phylogenetic analysis. The motif analysis revealed that CaTCPs in the same class exhibit similar numbers and distribution of motifs. This study predicted that 37 previously reported miRNAs target 19 CaTCPs. The expression levels of CaTCPs varied in various tissues and growth stages. Specifically, CaTCP16, a member of Class II (CIN), exhibited significantly high expression in flowers. Class I CaTCPs exhibited high expression levels in leaves, while Class II CaTCPs showed high expression in lateral branches, especially in the CYC/TB1 subclass. The expression profile suggests that CaTCPs play specific roles in developmental processes of pepper. This study provide a theoretical basis that will assist in further functional validation of the CaTCPs.

Minor corrections are

1.     In many occasions chromosome and its number (chromosome5) are joined and some places it is written as chromosome 5

2.     Line 264-266, not clear. Reframe the sentence

3.     Line 337-338, not clear. Reframe the sentence

4.     Line 358, the results showed that CaTCP7, CaTCP12 and CaTCP16 were

5.     Figure 6 fonts are visible. Increase the font size. Network diagram of the regulatory linkages among the anticipated miRNAs and putative CaTCPs. Green octagon colors correspond to miRNAs, and gray rectangle represent CaTCPs.

Round 2

Reviewer 2 Report

Comments and Suggestions for Authors

Below are my comments for the previous version.

1.       MGEA X and HMMER 3.1 are old software. Analysis should be performed with the latest MEGA 11 or HMMER 3.4.

2.       Supplementary files have not been uploaded. It is referenced in the text, but the data cannot be verified.

I understand the response to comment 1. If you have confirmed it with MEGA11 and HMMER 3.4, I think it would be better to include the figure in the main text. By the way, it's "MEGA", not "Mega".

Regarding comment 2, I checked the supplementary figure and table. I think it would be better to include lengend in the supplementary figure.
